# Prevention and Eradication of Biofilm by Dendrimers: A Possibility Still Little Explored

**DOI:** 10.3390/pharmaceutics14102016

**Published:** 2022-09-22

**Authors:** Silvana Alfei, Debora Caviglia

**Affiliations:** 1Department of Pharmacy, University of Genoa, Viale Cembrano 4, 16148 Genoa, Italy; 2Department of Surgical Sciences and Integrated Diagnostics (DISC), University of Genoa, Viale Benedetto XV 6, 16132 Genova, Italy

**Keywords:** multidrug resistance, bacterial BFs, fungi BFs, *P. aeruginosa* BFs, cationic antimicrobial agents, cationic dendrimers, antibiofilm agents

## Abstract

Multidrug resistance (MDR) among pathogens and the associated infections represent an escalating global public health problem that translates into raised mortality and healthcare costs. MDR bacteria, with both intrinsic abilities to resist antibiotics treatments and capabilities to transmit genetic material coding for further resistance to other bacteria, dramatically decrease the number of available effective antibiotics, especially in nosocomial environments. Moreover, the capability of several bacterial species to form biofilms (BFs) is an added alarming mechanism through which resistance develops. BF, made of bacterial communities organized and incorporated into an extracellular polymeric matrix, self-produced by bacteria, provides protection from the antibiotics’ action, resulting in the antibiotic being ineffective. By adhering to living or abiotic surfaces present both in the environment and in the healthcare setting, BF causes the onset of difficult-to-eradicate infections, since it is difficult to prevent its formation and even more difficult to promote its disintegration. Inspired by natural antimicrobial peptides (NAMPs) acting as membrane disruptors, with a low tendency to develop resistance and demonstrated antibiofilm potentialities, cationic polymers and dendrimers, with similar or even higher potency than NAMPs and with low toxicity, have been developed, some of which have shown in vitro antibiofilm activity. Here, aiming to incite further development of new antibacterial agents capable of inhibiting BF formation and dispersing mature BF, we review all dendrimers developed to this end in the last fifteen years. The extension of the knowledge about these still little-explored materials could be a successful approach to find effective weapons for treating chronic infections and biomaterial-associated infections (BAIs) sustained by BF-producing MDR bacteria.

## 1. Introduction to Microbial Resistance

The incidence of microbial infections has expanded dramatically, mainly due to the increasing occurrence of resistance among diverse strains of bacteria. The resistance to drugs is described as the tolerance or insensitivity of a microbe to an antimicrobial drug despite earlier susceptibility to it [1,2]. Concerning drug resistance (DR), different definitions are used in the medical literature to characterize the different patterns of resistance found in healthcare-associated, antimicrobial-resistant bacteria. Particularly, bacteria can be multidrug-resistant (MDR), extensively drug-resistant (XDR), and pandrug-resistant (PDR) [3]. So, MDR bacteria are those that have acquired non-susceptibility to at least one antibiotic in three or more antimicrobial categories, XDR bacteria are defined as non-susceptible to at least one antibiotic in all but two or fewer antimicrobial categories, and PDR bacteria are those non-susceptible to all agents in all antimicrobial categories [3].

Resistant bacteria, fungi, viruses, and parasites are able to combat the attack of available drugs, which are no longer effective, thus resulting in the persistence and spreading of chronic infections. The development of MDR is a natural phenomenon; however, excessive use of antibiotics, among humans, animals, and plants, incessantly supports its expansion [4]. Additionally, the widespread increase in immunocompromised individuals, such as patients affected by HIV infection, diabetic patients, persons who have experienced organ transplantation, and severe burn patients, makes the human body an easy target for hospital-acquired infectious (HAIs), thus contributing to further spread of MDR pathogens. From studies on WHO reports concerning the rates of resistance in bacteria, such as *Escherichia coli, Klebsiella pneumoniae, Staphylococcus aureus, Streptococcus pneumoniae, Nontyphoidal Salmonella, Shigella* species, *Neisseria gonorrhoeae*, and *Mycobacterium tuberculosis,* several fungi, viruses, and parasites, against several different classes of antibiotics, it has emerged that they are responsible for severe infections such as urinary tract infections (UTIs), pneumonia, and bloodstream infections (BSIs) [5,6]. The most common MDR microbes, tolerated drugs, and related hospital-acquired infections are reported in Table 1.

Antifungal drugs available for the treatment of chronic fungal infections are limited and resistance to drugs such as amphotericin B, ketoconazole, fluconazole, itraconazole, voriconazole, flucytosine, and echinocandins was found in isolates of *Candida* spp., *Aspergillus* spp., *C. neoformans, Trichosporon beigelii, Scopulariopsis* spp., and *Pseudallescheria boydii* [7,8,9,10]. Additionally, antiviral resistance has become a serious matter of concern in immunocompromised patients, including immunosuppressed transplant receivers and oncology patients infected by CMV, HSV, VZV, HIV, influenza A virus, hepatitis C (HCV), or HBV [11,12,13,14,15]. Parasitic MDR has been observed in isolates of *Plasmodia, Leishmania, Entamoeba, T. vaginalis*, *Schistosomes* [16,17,18,19,20,21,22,26,27], and *T. gondii* [23,24,25] against drugs such as chloroquine, pyrimethamine, artemisinin, pentavalent antimonials, miltefosine, paromomycin, and amphotericin B, in addition to atovaquone and sulfadiazine. A leading example of a parasite with a high tendency to develop MDR is *P. falciparum*, which is the cause of malaria. Other protozoan MDR parasites are *Entamoeba* spp., which causes amoebiasis, a major public health threat in many tropical and subtropical countries; and *Schitosomes* which is responsible for schistosomiasis, considered a global health concern similar to malaria and other chronic diseases [27].

## 2. Introduction to Biofilm (BF)

BFs are organized communities of microorganisms, including sessile cells, persistent cells, and dormant cells, which, during BF development, obtain physiological characteristics differentiating them from planktonic cells. Differently from the latter, which live suspended in a medium, sessile cells are cells attached to surfaces, forming highly coordinated microcolonies that lack motility [28]. They are incorporated into an extracellular polymeric matrix, self-produced by the bacteria, which adheres to living or abiotic surfaces present both in the environment and the healthcare environment [28]. The formation of BF by bacteria is a valid adaptation strategy that protects them from hostile environments, including host defenses, disinfectants, and antibiotics [28]. BF biomass consists of extracellular polymeric substances (EPSs), including a combination of enzymatic proteins; polysaccharides, such as cellulose, polyglucosamine (PGA), and exopolysaccharides; extracellular DNA (eDNA); and cationic and anionic glycoproteins and glycolipids, which allow real communication between the bacteria and stabilize the three-dimensional structure of the BF itself [29,30]. Table 2 summarizes the general composition of BFs.

The nutrients within the matrix are used by bacteria, which in turn secrete enzymes capable of changing the composition of EPSs in response to changes in the nutrient availability, while water is retained through H bonds with hydrophilic polysaccharides [29,30]. In Gram-negative species, the formation of BF begins with the anchoring of planktonic cells to a surface by pili and bacterial flagella [31,32]. Differently, in Gram-positive species, anchoring occurs through surface proteins [33]. Following adhesion, the bacteria begin to proliferate and form microcolonies and produce the extracellular matrix, thus allowing the integrity of the BF, whose matrix, in addition to exopolysaccharides, proteins, and DNA, also includes both bacterial and host lytic products [34,35,36]. Following the maturation of BF, the dispersion phase occurs, in which the secretion of enzymes (enzymes of the degradation of polysaccharides, proteases, nucleases, etc.) responsible for the disintegration of the matrix is observed, allowing other bacteria to leave BF, colonize new surfaces, and then form new BFs, leading to the spread of infection [34,35,36]. As an example, the phases of bacterial BF development are shown in Figure 1.

The genesis of BF is related to quorum sensing (QS), a mechanism that allows intracellular communication and regulates gene expression in response to cell density [30]. QS occurs in both Gram-positive and Gram-negative bacteria, but the self-inductors that are produced, and/or the signal molecules that allow cell-to-cell communication are different. In general, QS allows bacteria to begin, synchronize, and create a synergy that allows the architecture of BF to be retained and the creation of an environment within it that is conducive to the survival of pathogens [37,38]. There are many abiotic or living surfaces, including tissues, industrial surfaces, medical devices, dental materials, and contact lenses, on which pathogens can form BF [37,38].

Note that BF succeeds in cancelling antibiotic activity because drugs are incapable of diffusing into its complex structure and reaching pathogens [39]. Additionally, several overcoming factors, including a high cell density, an increase in the number of resistant mutants, molecular exchanges, release of substances, an increase in the expression of efflux pumps, altered bacterial growth rates, different gene expression, and persistent and dormant cells, make it difficult for antibiotics to counteract BF production [29,40]. The formation of BFs occurs mainly in chronic infections, characterized by the persistence of etiological agents, while acute infections are generally sustained by planktonic bacteria [41,42,43,44]. BFs are involved in over 60% of chronic wound infections, which can be colonized by a single or several bacterial species [45]. In this scenario, the major bacteria involved are *S. aureus* and *P. aeruginosa* [46,47]. In this regard, chronic respiratory infections sustained by *P. aeruginosa* are one of the main causes of mortality in patients suffering from cystic fibrosis. Unfortunately, evidence attests that 80% of the patients affected by cystic fibrosis are likely to develop chronic respiratory infections [48]. Furthermore, *P. aeruginosa*, following the formation of BF on the surface of the endotracheal tubes, is responsible for the development of pneumonia in intubated patients [49]. Eradication of chronic infection is hindered by the growth mode of BF, the intrinsic resistance of *P. aeruginosa* to antibiotics, and a high presence of hypermutable strains [50,51]. Regarding *S. aureus*, it is mainly responsible for the formation of BF on medical devices and the onset of biomaterial-associated infections (BAIs). In BAIs, the development of the infection depends on the type of implant and the length of time the implant is in the patient. The adhesion of pathogens to permanent medical devices is favored by fibronectin and fibrinogen, which act as adhesion mediators for staphylococci [52]. Concerning fungi, *Candida* species, which are often found in the normal microbiota of humans, are commonly found on implanted biomaterials and medical instrumentation [53,54,55], where they can form BF structure, thus being one of the main causes of catheter-related infections [5] and posing an important health risk for hospitalized patients [55,56,57]. BFs of *Candida* species have significant resistance to antifungal agents and the ability to withstand harsh conditions, and BF cells are capable of evading the host’s immune response [58], thus representing a significant challenge for patients with immunodeficiency or medical-implanted devices [59]. Moreover, since *Candida* is a eukaryotic microorganism, similar to human cells, the successful design of new antifungal compounds is limited by its poor selectivity and tendency to present high cytotoxic levels [60]. Collectively, there is an urgent need to find alternative options to prevent and control, among other microorganisms, the formation of *P. aeruginosa, S. aureus*, and *C. albicans* BF and BF-related infections. In recent years, various antibacterial surfaces or antibacterial coatings for surfaces have been developed to attempt to counteract the adhesion and colonization of surfaces by pathogens producing BFs. In general, antibacterial surfaces can be divided into antifouling or bactericide, depending on the effects they are able to exert on the biological systems they encounter [61].

The antifouling activity of surfaces is achieved by the use of hydrophilic polymers or zwitterionic polymers while bactericidal activity usually leads to the inactivation of pathogens that adhere to the surface or are in suspension close to surfaces. In the latter case, depending on the pathogen-killing mechanism, there are two types of bactericidal surfaces: those that kill bacteria by contact, and those that kill bacteria by releasing bactericidal agents [61]. Unfortunately, currently, no antibacterial surface is capable of completely hampering the formation of BF, by either inhibiting its formation or causing its degradation. To address this need, researchers have developed novel methods for designing novel functional antibacterial surfaces with joint bactericidal and antifouling properties [62]. Interestingly, dendrimer compounds have appeared as interesting alternatives to conventional drugs in biomedicine and could be a new therapeutic approach to combat the infections caused by MDR pathogens, including those producing BFs [63]. Dendrimer systems have well-defined and monodispersed structures and are being widely studied for various biomedical applications, such as drug delivery systems, antivirals, and magnetic resonance imaging contrast agents, in addition to antibacterial and antitumor drugs [63,64,65,66,67,68,69]. Concerning cationic dendrimers, properties such as mono-dispersity, high water solubility, and multivalency allow them to act as powerful therapeutic agents, with significant potential for clinical applications as antibacterial agents [70,71,72]. Particularly, multivalency provides dendrimers with several functional groups on the surface capable of detrimentally interacting with bacterial membranes, causing disruption and killing of pathogens [70,71,72]. In this regard, cationic dendritic molecules have been studied against different planktonic cells [73,74,75,76] and BFs [62,77], showing a low tendency to induce antibiotic resistance in bacteria [78]. Furthermore, the synergistic combination of these structures with commercial drugs can improve the drug’s solubility and antibacterial activity [75,79,80] more than one therapeutic function and can allow a reduction in the administered doses, reducing the side effects of the drugs.

In this scenario, the main scope of this paper is to incite further development of new antibacterial agents capable of inhibiting BF formation and, hopefully, destroying mature BF. To this end, we first summarize the useful information concerning microbial resistance, BF, and specifically relating to BFs by *P. aeruginosa*, which is one of the most clinically relevant opportunistic bacteria producing BF on medical devices, responsible for untreatable infections. Secondly, we review all the dendrimers developed in the last fifteen years that have shown an ability to counteract this worrying form of bacterial resistance. We are confident that the extension of the knowledge about this still little-explored nanomaterial is a successful approach to find effective weapons for treating chronic infections and biomaterial-associated infections (BAIs) sustained by BFs produced by MDR bacteria.

### 2.1. BFs by P. aeruginosa

The formation of BF is an unceasing cycle, in which, as predicted in Section 2, organized communities of microorganisms are trapped in a matrix of EPSs that holds microbial cells together and attached to a surface [63]. BFs are symbolically called a “city of microbes” in which EPSs, representing 85% of the total BF biomass, constitute the “house of the BF cells”. BF cells handle 65–80% of all microbial infections. BFs have become a major issue in the medical sector since BFs are also a major cause of chronic infections due to their high resistance to antibiotics and the host immune response. Note that, in natural environments, BFs rarely exist as mono-species BF while interspecies interactions affect many genetic and phenotypic attributes in multispecies BFs. Consequently, it is crucial to focus mainly on the study of multispecies BFs to understand BFs better. Nevertheless, to date, BF research has mostly been limited to the study of mono-species BFs. In this regard, *P. aeruginosa* is a pathogenic opportunistic bacterium that has been widely studied for its high incidence in clinical settings and its ability to form strong BFs. *P. aeruginosa* is an opportunistic pathogen capable of adapting to various environments and developing resistance against multiple classes of antibiotics. The distinct main developmental stages of *P. aeruginosa* BF are shown in Figure 2.

#### 2.1.1. Attachment of *P. aeruginosa* BFs

Although early studies suggested that simple chemical bonds such as Van der Waals forces contribute to the first attachment of bacteria of the BF-forming *Pseudomonas* genus, it has been shown that much more complex events and bacterial structures are involved in the early stage of BF development. Adhesins, type IV pili, and lipopolysaccharide (LPS) promote the attachment of bacteria to surfaces, and these bacterial structures are in turn specifically regulated by environmental signals [81,82]. Additionally, recent studies proved that the origins of BF formation occur simultaneously with an improvement in the levels of c-di-GMP, which is a second messenger [83,84,85,86,87] that activates the production of adhesins and various ECM products [83,87]. Moreover, BF formation is also regulated by sRNAs, which in turn control Psl, Pel, and alginate production, which are extracellular polysaccharides (EPSs) implicated in BF development and the motile-to-sessile switch of *P. aeruginosa* cells [88,89]. Particularly, Pel and Psl are produced by non-mucoid strains of *P. aeruginosa*, mainly playing an important role in surface attachment for most isolates, and there is significant strain-to-strain variability in the contribution of Pel and Psl to mature the BF structure [90].

#### 2.1.2. Maturation of *P. aeruginosa* BFs

After adhering reversibly to surfaces or each other, bacterial cells undergo the switch from reversible to irreversible attachment. *P. aeruginosa* bacteria undergo a series of changes to adapt to the new mode of life, thus shifting from the status of planktonic cells (free-living cells) to that of sessile cells (not motile attached cells). They form a more structured architecture, termed microcolonies, in which bacteria are highly coordinated and start working together to produce ECMs and build structures and water channels. These microcolonies develop further into extensive three-dimensional mushroom-like structures, a hallmark of BF maturation. While BF matures, *P. aeruginosa* bacteria go through physiological modifications, thus becoming more resistant to environmental stresses and antibiotics. All this machinery is governed by a signaling system called quorum sensing (QS) [63].

#### 2.1.3. Detachment of *P. aeruginosa* BFs

The costs associated with BF growth make it pivotal that bacteria have mechanisms to separate from BFs and return to planktonic life. This process is referred to as detachment or dispersion. Detachment is the final stage of BF development, essential to the creation of new BFs in new niches. Detachment can occur by several different mechanisms, such as sloughing, erosion, and seed dispersal [82,91,92]. While sloughing and erosion are passive detachments and are mediated by shear stress [82,91], seed dispersal consists in the active disengagement mechanism of *P. aeruginosa* BFs, during which single planktonic cells or microcolonies are released by the center of *P. aeruginosa* BFs, leaving an empty cavity (central hollowing) [91]. This mechanism includes the degradation of ECM and the autolysis of a BF subpopulation of cells, including dormant and persistent cells. Recently, it was reported that endonuclease EndA is needed for the dispersion of an existing BF via eDNA degradation. BF dispersion can also be promoted by environmental signals, such as variations in nutrients, the availability of oxygen, nitric oxide (NO), pH, and various chemicals. Collectively, signals that cause decreases in the levels of pyoverdine and intracellular c-di-GMP and increases in flagella production induce dispersal. Moreover, metal chelators and compounds such as cis-2-decenoic acid, anthranilate, or other surfactants can induce BF dispersal [93,94,95].

Figure 3 shows a BF in the dispersal phase and reports the main factors, cues, and signals that induce BF detachment.

Furthermore, in vitro and in vivo experiments have shown that the cells liberated by BFs are extremely cytotoxic to macrophages, more sensitive to iron diminution, and significantly more virulent to nematode hosts than planktonic bacteria. Additionally, it was reported that dispersed bacteria derived from BFs treated with glycoside hydrolase rapidly induced fatal septicemia in a mouse chronic wound infection model.

#### 2.1.4. Important Characteristics of *P. aeruginosa* BF

Table 3 shows some important characteristics of BF produced by *P. aeruginosa*. Particularly, among the three identified types of EPSs in *P. aeruginosa* (Psl, Pel, and alginate) [96], Psl was named based on the locus of the polysaccharide synthesis, which was identified in 2004 [97,98]. In the late stage of BF maturation, Psl accumulates on the outside of structured BFs [99,100] but can also form a web of eDNA–Psl, thus supplying structural support to BF for its later dispersion. Interestingly, the eDNA–Psl interactions could increase the extent of *P. aeruginosa* in vivo by the use of neutrophil extracellular traps as a BF scaffold [101].

Among the most accredited hypothesis, it has been reported that e-DNA may be produced by active secretion, autolysis of bacteria, or release from small membrane vesicles [99,107]. Concerning the QS system, which is an intercellular communication system that allows bacteria to feel their own population density [63], it relies on small signaling molecules such as N-acyl-homoserine lactones (NAHSL) and the and autoinducer-2 (AI-2). Note that while the latter has also been found in Gram-positive bacteria, they use oligopeptides in place of lactones. Concerning the four types of QS systems in *P. aeruginosa*, while the las QS system is involved in the production of N-3-oxododecanoyl homoserine lactones (N-3-C12-HSL), the rhl QS system is implicated in the synthesis of N-butanoyl-L-homoserine lactone (C4-HSL). Additionally, the las and rhl QS systems control much gene expression, such as the production of elastase, protease, rhamnolipids, and other factors of virulence. Moreover, the PQS system controls the release of eDNA in BF formation and the production of membrane vesicles [119,120] and affects many other metabolic processes in *P. aeruginosa*, including iron chelation, redox homeostasis, the production of elastase and rhamnolipids, the formation of membrane vesicles, etc. [117,121]. IQS is a recently discovered QS system, which can integrate environmental stress cues into QS. These QS systems have hierarchical relationships among them, with the las system being in the highest position in the QS system while the rhl system is at the lowest level. Nevertheless, each QS system can also be initiated by environmental factors, including phosphate stress, starvation, low oxygen, low iron, and several host-derived factors [117,122,123,124,125].

#### 2.1.5. *P. aeruginosa* BF-Associated Infections

Infections associated to BF produced by *P. aeruginosa* can be classified into two categories. The first category is represented by BF infections due to indwelling medical devices, such as central venous catheters, urinary catheters, prosthetic couplings, peritoneal dialysis catheters, pacemakers, contact lenses, and intrauterine devices. The second class consists of direct BF infections in host tissues, such as chronic pneumonia in patients with CF, chronic otitis media, endocarditis, chronic osteomyelitis, chronic prostatitis, palindromic urinary tract infections (UTIs), and gingivitis [99]. The major issue associated with the development of BF infections in diverse medical environments consists in their significant resistance against most antibiotics and other disinfectants. The specific characteristics of BFs hamper the diffusion of drugs, which become uncapable of reaching bacteria, thus nullifying the antibiotic activity [63]. To counteract bacteria capable of forming BFs, an antimicrobial agent must prevail over several additional obstacles, including an increased number of resistant mutants, high cell density, molecular exchanges, substance delivery, efflux pumps, persistent and dormant cells, altered bacteria growth rate, and different gene expression [63]. In this regard, Table 4 reports the mechanisms through which BFs hamper the activity of antibiotics, with examples of BF-producing microorganisms and the related inactivated antibiotics.

*P. aeruginosa* can use the mechanisms shown in Table 4 to infect and occupy various locations of the human body. *P. aeruginosa* is notorious for causing pneumonia in the lungs of CF patients, thus being the primary cause of death of CF patients [138]. *P. aeruginosa* BFs in the CF lung consist of small aggregates wrapped in EPSs. BFs induce inflammation of the infected lung by means of enrolling polymorphonuclear leukocytes. BF allows bacteria to survive inflammation and aggressive antibiotic treatments, thus causing persistent infection. The chronic inflammatory response against BF infections triggers tissue damage and leads to lung failure [139]. Otitis media is another infection sustained by *P. aeruginosa* BF. Particularly, it is the infection of the middle ear. Very common among children, it can cause serious inflammation that may lead to conductive hearing loss [140]. In this case, BF consists of small microcolonies of less than 100 bacteria. Additionally, *P. aeruginosa* can cause chronic bacterial prostatitis, which is an infection of the prostate gland, representing the major cause of relapsed UTIs in men, in which microcolonies of *P. aeruginosa* are associated with the ductal wall of the prostate duct and cause the disease [139,141]. One of the major complications of *P. aeruginosa* BF infection is represented by chronic wound infections. Chronic wounds are normally correlated with vascular abnormalities such as decubitus ulcers, ischemic injuries, diabetic foot ulcers, and venous leg ulcers [140,141]. Since the skin barrier is compromised, these chronic wounds generate suitable environments for bacteria attachment and colony formation. Microbial infections in chronic wounds are multispecies infections, consisting of both aerobic and anaerobic bacteria. Among the isolated bacteria from chronic wounds, *P. aeruginosa* and *S. aureus* are the most common ones [142,143]. *P. aeruginosa* exists in BFs in wounds, located in a deeper part of wounds than to *S. aureus*. Furthermore, chronic wounds with *P. aeruginosa* infection tend to be larger, more inflamed, and slower to recover [140,142]. This could be due to characteristics typical of *P. aeruginosa* BF, such as type IV pili and flagella-mediated motility, in addition to the production of virulence factors that protect the bacteria from host defense systems [139]. Another very important class of *P. aeruginosa* BF infections includes infections on medically implanted devices. In this regard, *P. aeruginosa* strains are often isolated from infected urinary catheters, intravascular catheters, artificial joints, and cochlear implants [139]. BFs have been isolated from almost all medical device-related infections and are very difficult to remove. These infections are at high risk of progression to systemic infections. Thus far, the only treatment of BF infections on medical devices is removal of the device.

## 3. Prevention and/or Eradication of BF by Dendrimers: A Possibility Still Little Explored

Although cationic dendrimers have been extensively studied as antibacterial agents with high potentiality, as shown by a search of Scopus using “cationic dendrimers” and “biofilm” as keywords, in the last 15 years (2007–2022), only 29 documents concerning this topic have been reported in the literature (Figure 4).

This scenario shows that, although good premises exist, the possibility of using dendrimers to prevent the formation of BF or even to promote the breakdown of mature BF is, to date, still little explored. Unfortunately, the research in this field, instead of increasing, was shut down and after 2016, only five jobs were reported. This phenomenon highlights a diminished interest of a significant number of experts in developing new cationic dendrimers active against BF, further underlining the difficulty in being successful in such a challenge compared to being successful in developing new cationic materials, including dendrimers with bactericidal activity against bacteria with other types of resistances.

### 3.1. Antibacterial Cationic Macromolecules

Currently, there are few molecules under clinical development that are active against MDR pathogens and/or BF producers. Concerning the main agents in clinical development (Phase III) in 2020, only caspofungin, which is an antifungal drug, has been proven to inhibit the synthesis of polysaccharide components of the bacterial BF of *S. aureus* [144]. Concerning nanoparticles, some research groups are currently studying polymeric lipid nanoparticles, involving the conjugation of rhamnolipids (biosurfactants secreted by the pathogen *P. aeruginosa*) and polymer nanoparticles made of clarithromycin encapsulated in a polymeric core of chitosan. By the same principle, rhamnolipid-coated silver and iron oxide NPs have been developed, which have been shown to be effective in eradicating *S. aureus* and *P. aeruginosa* BFs [144]. Therefore, to counteract the phenomenon of drug resistance and meet the urgent need for new antibacterial agents that are also active in infections sustained by BF cells, the search for alternative therapeutic strategies capable of acting under mechanisms different from those of conventional drugs is a daily challenge of researchers in this field. In this regard, various natural cationic molecules called antimicrobial peptides (NAMPs) have shown significant capabilities to limit or inhibit bacterial growth and represent excellent candidates for replacing current antibiotics that are no longer effective. The amphiphilic structure and the net positive charge of NAMPs are the two most important requisites necessary for antibacterial effects and decide their mechanism of action [63,145,146,147]. It has, in fact, been shown that NAMPs’ action causes the death of the bacterial cell in a non-specific way from the outside, without having to enter the bacterial cell and interact with its metabolic processes, which is likely to genetically mute the conferring of resistance [63,145]. The action of NAMPs on bacteria is rapid, causes depolarization and destabilization of the membranes, and generates pores that gradually lead to an increase in their permeability, with consequent leakage of essential cations and other cytoplasmic materials and cell death. Since it is not connected to mutable bacterial processes, this mechanism of action generally allows the cationic peptides to induce a lower development of resistance in the target cell compared to traditional antibiotics [63,145]. Although the main target of NAMPs is bacterial membranes, it has been reported that some of these cationic antimicrobial peptides, following the increased permeability of bacterial envelopes, could also enter the cell and irreversibly damage molecules, such as DNA, RNA, and enzymes, thus leading to an improvement in their original therapeutic efficacy [63,146,147,148,149]. Unfortunately, the in vivo application of NAMPs is hampered by their early inactivation by peptidases, their high hemolytic toxicity, and high costs of production. Based on this, in the last decades, inspired by NAMPs, new cationic antibacterial macromolecules, such as polymers, copolymers, and dendrimers, have been developed, with proven interesting antibacterial effects. Such macromolecules, unlike small antibacterial compounds, in addition to possess multivalence, have other several advantages, including a limited residual toxicity, greater long-term activity, greater chemical stability, and lesser tendency to develop resistance [63,147,150]. Among polymers, cationic dendrimers (CDs) are nonpareil, nano-sized, hyper-branched cationic macromolecules with a tree-like architecture and a unique spherical shape [63,64,65,66,67,68,69,70,71,72]. Dendrimers, which have been extensively applied in biomedicine for years, have recently been found to act as potent antibacterial agents and also exploitable for coating surfaces [62,63,64,68,69,70,71,72].

#### Cationic Dendrimers

Dendrimers were first synthesized in the mid-1980s, but their use as antibacterial agents, mimicking the action of NAMPs for use as drugs, surface coating agents, or drug-delivery systems, has only recently been recognized [62,63,64,68,69,70,71,72]. In the last decade, dendrimers have mainly been synthesized for the treatment of infections caused by MDR pathogens and some of these have been shown to have antibiofilm action [151,152]. The most used and studied commercial dendrimers as antibacterial agents are poly(amidoamine) (PAMAM) and polypropylene imines (PPIs), which have shown wide activity in vitro; however, unless they are properly modified to improve their biodegradability, reduce their susceptibility to opsonization and toxicity against eukaryotic cells, and fast clearance, they are not clinically applicable [70,71,72]. Moreover, polymers such as linear and branched poly (ethylene imine)s (PEIs), in addition to requiring minor costs for production, although with less perfect architectures than PAMAMs and PPIs, are known for their ability to enter cells or permeabilize cell membranes. Accordingly, while a large number of studies have focused on the antibacterial activity of water-soluble PEI derivatives containing quaternized ammonium salt groups with long alkyl or aromatic groups, others reported the application of water-insoluble hydrophobic PEIs, including nanoparticles, as antibacterial coatings [153]. To improve and therefore obtain CDs with better characteristics, particular attention was paid to the synthesis of biodegradable polyester-based dendrimer scaffolds, peripherally modified with suitable amino acids, thus achieving shells that are highly cationic and conferring the obtained dendrimer NPs’ potent antibacterial properties [63,70,71,72,154,155]. In the worrying scenario where the weapons used to counteract MDR bacteria and above all infections sustained by BF-producing pathogens are dramatically decreased or non-existent, cationic dendrimer NPs could represent new valid promising tools for fighting MDR pathogens and also for the treatment of sustained BAIs of MDR BF-producing bacteria. The advantages associated with the use of cationic material with dendrimer structures mainly depend on their high multivalency deriving from their tree-like and generational structure, which allows a large abundance of active cationic moieties, thus greatly improving their antibacterial potency [63]. Aiming to inspire the realization of further synthetic strategies for developing new antibacterial dendrimers capable of inhibiting the first training and/or destroying mature BFs, we review the most recent investigations carried out to design and develop new antibiofilm dendrimer agents and the related results.

### 3.2. Recently Reported Case Studies 

In the following part of the present paper, we report the dendrimer materials developed in the last fifteen years that were tested for their effects in preventing or limiting the development of fungi and bacterial BF, and causing its dispersal. In this regard, Table 5 shows the structures representative of the main class of dendrimers engineered.

A research line started in the first decade of the 2000s concerned the synthesis and screening of libraries of dendrimer compounds with C-fucosyl and galactosyl residues with an affinity for fucose- and galactose-specific lectins. The scope was to detect potent fucose and/or galactose inhibitors of lectins LecA and LecB produced by *P. aeruginosa*, which are implicated in tissue binding and BF formation. In this regard, regarding the results obtained the previous year when a very large library of peptide dendrimers was screened to assess their affinity for LecB, the most potent LecB-ligands identified as dendrimers (FD2 (C-Fuc-LysProLeu)4(LysPheLysIle)2 LysHisIleNH2 (IC50 = 0.14 mM by ELLA) and PA8 (OFuc-LysAlaAsp)4(LysSerGlyAla)2 LysHisIleNH2 (IC50 = 0.11 mM by ELLA)) were tested on BF of *P. aeruginosa* by Johansson et al. in 2008 [156]. Particularly, FD2 led to complete inhibition of *P. aeruginosa* BF formation (IC_50_ = 10 mM) and induced complete dispersion of mature BFs in the wild-type strain and in numerous clinical *P. aeruginosa* isolates, thus indicating that LecB inhibition by high-affinity multivalent ligands such as FD2 could correspond to a curative approach against BF-associated chronic infection sustained by *P. aeruginosa* [156]. Subsequently, Kolomiets et al. (2009), building on promising results previously published, prepared 10 tetravalent and 3 octavalent water-soluble C-fucosyl peptide dendrimers by solid-phase peptide synthesis (SPPS). By determining the relative affinities of these ligands to LecB using an enzyme-linked lectin assay (ELLA), strong binding of up to a 440-fold enhancement in potency over fucose was observed for the octavalent cationic dendrimer 2G3, mainly due to the multivalency of the dendrimer. Collectively, due to the potent binding and inhibition action on LecB by 2G3 and the versatility and reliability of SPPS to produce tunable multivalent fucosylated peptide dendrimer ligands, the dendrimers reported by Kolomiets et al. seemed excellent candidates for the development of polyvalent inhibitors of *P. aeruginosa* adhesion and BF for use in therapy to prevent BF-associated infections [157]. Antibacterial dendrimer peptides, acting in this case as membrane disruptors due to their positive charge ((RW) _4D_), were prepared by Hou et al. (2009), which were effective in preventing the formation of *E. coli* BFs, hindering their development and destroying mature forms [158]. In 2011, Johansson et al. carried out research on a potent LecB inhibitor active in the prevention and dispersal of BF by *P. aeruginosa*, proposing a strategy to limit the early in vivo deactivation of FD2 by proteases. In this regard, he synthetized the C-fucosyl peptide dendrimer D-FD2 (CFuc-lys-pro-leu)4(Lys-phe-lys-leu)2Lys-his-leu-NH2 using D-amino acids and obtained the stereoisomer of FD2 [159]. Although D-FD2 showed a lower affinity for the fucose-specific lectin LecB, it was active as a BF inhibitor and, in addition, showed high stability towards proteolysis with pure proteases and in human serum, thus being conceivable for future clinical applications. In summary, the peptide dendrimer D-FD2 was prepared by substituting D-amino acids for L-amino acids in the branches of the dendrimer FD2. Collectively, the exchanged geometry altered the binding affinity to LecB while retaining the *P. aeruginosa* BF inhibitory properties and supplying complete resistance to proteolysis [159]. Establishing that BF formation by *P. aeruginosa* is mediated in part by the galactose-specific lectin LecA (PA-IL) and the fucose-specific lectin LecB (PA-IIL), and understanding that the glycoconjugate–lectin interaction is a key feature in developing potent BF inhibitors, Kadam et al., deepening their research in the field, in 2011, reported the first case of *P. aeruginosa* BF inhibition by the β-phenylgalactosyl peptide dendrimer GalAG2 multivalent ligand, this time targeting the galactose-specific lectin LecA [160]. Since it was observed that hydrophobic groups in the sugar anomeric point improved the affinity of galactosides to LecA, the acetyl-protected 4-carboxyphenyl β-galactoside (GalA) was connected to the peptide dendrimer [160]. To investigate the effect of the sugar-dendrimer linker on the binding affinity, carboxypropyl β-thiogalactoside (GalB) was also introduced as the last building block in solid-phase peptide synthesis to give the dendrimers GalAG1-G2 and GalBG1-G2, and the linear peptides GalAG0 and GalBG0 [160]. Based on the reported results, the strongest binding was detected with the second-generation glycopeptide dendrimer GalAG2. Interestingly, both the GalAG2 and GalBG2 dendrimers exhibited potent BF inhibition while the G1 equivalents were much less active and the G0 analogs were ineffective [160]. In the same year (2011), Chen et al. assessed the activity of a dendrimer peptide with various numbers of Trp/Arg repeated residues ((RW)_4D_), which was previously found to have antibacterial effects by Hou et al. [158], against planktonic persister cells and BF-associated persister cells of *E. coli* HM22 [161]. The results were then compared with those obtained using three linear synthetic AMPs. Overall, (RW)_4D_ exhibited potent activities in eradicating BF cells and associated antibiotic tolerance in a dose-dependent manner [161]. Particularly, after treatment for 1 h with 20, 40, and 80 μM (RW)_4D_, the total number of viable BF cells was reduced by 77.1%, 98.8%, and 99.3%, respectively, compared to the untreated control (*p* < 0.0001). The tolerance to ampicillin was also reduced by 57.2%, 96.9%, and 99.1%, respectively (*p* = 0.0012). Furthermore, no viable cell was found after treatment of the BFs with 40 or 80 μM (RW)_4D_ followed by 5 μg/mL ofloxacin, suggesting that all persister cells were eliminated [161]. On these early results, (RW)_4D_ may be a pioneer compound for the development of new therapeutics to treat BF-associated infections that is also able to improve the action of antibiotics, including ampicillin and ofloxacin. Wang et al. (2011) presented a report on the antibacterial activity and cytotoxicity of three types of titanium-based substrates with and without calcium phosphate coatings on which poly(amidoamine) (PAMAM) dendrimers were immobilized (named Ti-S(CaPO_4_)060-PEGPAMAMs and Ti-S-060-PEGPAMAMs in the table) [162]. The utilized amino-terminated PAMAM dendrimers were modified with various percentages (0–60%) of PEG to obtain strong absorption on the titanium-based substrates and the formation of dendrimer films. The obtained films effectively inhibited colonization by *P. aeruginosa* (strain PAO1) and, although to a lesser extent, *S. aureus*. The antibacterial activity of the films was maintained even after storage of the samples in PBS for up to 30 days [162]. In addition, the dendrimer-based films showed low cytotoxicity to human bone mesenchymal stem cells (hMSCs) and did not alter the osteoblast gene expression promoted by the calcium phosphate coating [162]. The next year (2012), Scorciapino et al., using an alternation of hydrophilic and lipophilic amino acids, synthetized a lipodimeric dendrimer (SB056), which demonstrated an antibacterial effect by acting as a membrane disruptor [163]. The antibacterial activity of SB056 was similar to that of colistin e polimixin B against isolated MDR Gram-negative species while SB056 showed poor activity against strains of Gram-positive species. Interestingly, SB056 showed good antibiofilm effects against BFs produced by *S. epidermidis* and *P. aeruginosa* [163], thus being a future therapeutic weapon against BF-associated infections. The group of Lu et al., in 2013, recovered PAMAM-based dendrimers capable of releasing NO to develop new antibacterial agents, which are hopefully also effective against BFs [164]. Particularly, a series of amphiphilic PAMAM dendrimers were synthesized by a ring-opening reaction between the primary amine groups on the dendrimers and propylene oxide (PO), 1,2-epoxy-9-decene (ED), or a ratio of the two [164]. The obtained PAMAM-based dendrimers with different functionalities were then reacted with NO at 10 atm to produce N-diazeniumdiolate-modified scaffolds with a total loading of NO of ~1 μmol/mg [164]. In sight of a future clinical application of such materials, structure–activity relationship (SAR) studies were carried out to evaluate the bactericidal efficacy of the prepared NO-releasing delivery systems against established BFs by *P. aeruginosa* as a function of the dendrimer exterior’s hydrophobicity (i.e., ratio of PO/ED), size (i.e., generation), and NO release [164]. It was revealed that both the size and exterior functionalization of the dendrimer were pivotal for dendrimer–bacteria interaction, the NO delivery efficiency, bacteria membrane disruption, migration within BF, and toxicity to mammalian cells [164]. In this regard, the best PO/ED ratios for BF eradication with minimal toxicity against L929 mouse fibroblast cells were 7:3 and 5:5 [164]. The inhibition of BF formation by blocking the action of LecA and LecB lectins from *P. aeruginosa* was also investigated by Reymond et al. [165]. So, the four glycopeptide dendrimers that were synthesized showed high affinity for the lectins and were efficient in both blocking *P. aeruginosa* BF formation and inducing its dispersal in vitro [165]. As mimics of natural cationic amphiphilic peptides (NAMPs) with antifungal activity, eight peptide dendrimers were designed and evaluated for their anti-*Candida* spp. effects against both the wild-type strains and mutants by Zielinska et al. in 2015 [166]. Among the synthetized macromolecules, dendrimer **14**, containing four tryptophan (Trp) residues and a dodecyl tail, and dendrimer **9**, decorated with four N-methylated Trp, was shown to inhibit 100 and 99.7% of *Candida* growth at 16 µg/mL, respectively [166]. On these promising outcomes, **9** and **14** were designated for their evaluation against *C. albicans* mutants with deactivated biosynthesis of aspartic proteases, which are responsible for host tissue colonization and morphogenesis during BF formation (sessile model). According to the reported results, **14** affected *C. albicans* BF viability and the hyphal and cell wall morphology by membranolytic mechanisms [166]. By affecting the cellular apoptotic pathway and damaging the cell wall formation in mature BF, **14** may be a potential multifunctional antifungal template compound for the control of *C. albicans* chronic infections [166]. *C. albicans* was also the topic in the study of Lara et al. (2015), where novel antifungal strategies targeting BFs were explored [59]. Notably, using microwave-assisted techniques, nanosized spherical silver nanoparticles (AgNPs), aiming to investigate their potential biological applications, were prepared by Lara et al. without the addition of contaminants [59]. A potent dose-dependent inhibitory effect of AgNPs on BF formation of *C. albicans* was demonstrated (IC_50_ of 0.089 ppm). Additionally, AgNPs were shown to be efficient when tested against pre-formed *C. albicans* BFs, resulting in an IC_50_ of 0.48 ppm while the cytotoxicity assay resulted in a CC_50_ of 7.03 ppm. Using SEM and TEM analyses, it was evidenced that treatments with AgNPs caused outer cell wall damage, absence of true hyphae, filamentation inhibition, and membrane permeabilization [59]. The same year (2015), Bahar et al. developed the arginine-tryptophan-arginine 2D-24 dendrimer peptide, which was found to be effective against *P. aeruginosa* normal planktonic and persister cells, and against *P. aeruginosa* BF cells [152]. Particularly, the second-generation peptide dendrimer (2D-24), with residues of arginine and tryptophan, was active both on planktonic and BF cells, and against the MDR isolates of *P. aeruginosa* PAO1 (ATCC BAA-47) and PDO300 (mutans *mucA22* of *P. aeruginosa* PAO1) [152]. Collectively, 2D-24 displayed the same efficiency against both planktonic cells and BFs, thus establishing its capability to penetrate the BF’s mass and the alginate layer of mucoid isolates. Concentrations > 20 μM were sufficient to kill 80% of the planktonic and BF cells of PAO1 e PDO300 strains [152]. Further studies concerning combination therapies were carried out by Bahar et al., observing synergistic effects of 2D-24 with antibiotics such as ciprofloxacin (Cip), tobramycin (Tob), and carbenicillin (Car), all targeting the synthesis of DNA, proteins, and the cell wall [152]. The best result was obtained by combining 2D-24 with Tob. Additionally, in vitro studies on sheep erythrocytes and cocultures of PAO1 and human IB3-1 cells showed that 2D-24 exerted antibacterial effects at non-toxic concentrations on mammalian cells (25 µg/mL), thus providing encouraging evidence supporting the use of 2D-24 for chronic *P. aeruginosa*-borne infections [152]. Using a strategy similar to that used in 2013 by Lu et al. [164], Worley et al. described the synthesis of NO-releasing alkyl chain modified PAMAM dendrimers of the G1-G4 generations decorated with butyl or hexyl alkyl chains via a ring-opening reaction [167]. The resulting secondary amines were further reacted with *N*-diazenium-diolate NO donors, reaching an NO payload of ∼1.0 μmol/mg [167]. The bactericidal efficacy of these dendrimers was evaluated against BFs produced by isolates of *P. aeruginosa, S. aureus*, and MRSA. The anti-BF action of the dendrimers depended on the dendrimer generation, bacterial species, and alkyl chain length, with the most effective BF eradication occurring when antibacterial agents were capable of efficient BF infiltration. Regardless, the ability of dendrimers to release NO also helped the antibiofilm activity of those dendrimers incapable of effective BF penetration [167].

To better understand the binding mode of the tetravalent glycopeptide dendrimer (TGPD) GalAG2 to its target lectin LecA, the first crystal structures of LecA complexes with the divalent analog GalAG1 were obtained, revealing strong LecA binding and absence of lectin aggregation [168]. Secondly, a model of the chelate-bound GalAG27LecA complex was also obtained, rationalizing its unusually tight LecA binding (*K*_D_ = 2.5 nM) and aggregation by lectin cross-linking [168]. By evaluating the BF inhibition with divalent LecA inhibitor (GalAG1), it was indicated that lectin accumulation is necessary for BF inhibition by GalAG2, thus establishing that multivalent glycoclusters represent a unique opportunity to control *P. aeruginosa* BFs [168]. The next year (2016), the same group of Worley, this time headed by Becklund, studied in more detail the modification of (G1) (PAMAM) dendrimers with several alkyl epoxides to generate propyl-, butyl-, hexyl-, octyl-, and dodecyl-functionalized dendrimers [169]. The resultant secondary amines were treated with NO to form N-diazeniumdiolate NO donor-modified dendrimer scaffolds (total NO ∼1 μmol/mg) [169]. In this study, the bactericidal action of the NO-releasing dendrimers was tested against both planktonic and BF cells of *S. mutans*, with the greatest efficiency observed with the increase in the alkyl chain length and at lower pH. Particularly, the best bactericidal efficacy was obtained at pH 6.4, probably due to the increase in the cationic scaffold surface charge, which promoted dendrimer–bacteria association and the ensuing membrane damage. For shorter propyl and butyl chain modifications, however, the increased antibacterial action at pH 6.4 was due to the faster NO-release kinetics from proton-labile N-diazeniumdiolate NO donors. Octyl- and dodecyl-modified PAMAM dendrimers proved to be the most effective in both eradicating *S. mutans* BFs with the help of NO release and displaying mitigated cytotoxicity [169].

Inserting their studies in the research line concerning dendrimers that act as ligand inhibitors of LecA and/or LecB, Bergman et al. (2016) synthetized G3 and G4, analogous to GalAG2 and GalBG2, using a convergent synthetic technique and multivalent chloro-acetylcysteine (ClAc) thioether as the linker [170]. According to the reported results, G3 dendrimers showed an improved ability to link LecA with respect to the parent dendrimers G2 and were capable of exerting total inhibition of BF produced by P. aeruginosa, and causing its disaggregation [170]. The same year (2016), Michaud, Visini, Bergaman et al. focused on carrying out several changes in the structures of the earlier reported TGPDs acting as ligands of lectins LecB and/or LecA produced by *P. aeruginosa* to increase their BF inhibition activity [171]. As examples, they investigated heteroglycoclusters (Het1G2-Het8G2 and Het1G1Cys-Het8G1Cys), each having one pair of LecB-specific fucosyl groups and LecA-specific galactosyl groups, capable of simultaneously binding both lectins [171]. Interestingly, one of these ligands gave the first fully resolved crystal structure of the complex peptide dendrimer–LecB, providing a structural model for dendrimer–lectin interactions (PDB 5D2A) [171]. Furthermore, BF inhibition was increased by introducing additional cationic residues in these dendrimers. In a second approach, dendrimers with four copies of the natural LecB ligand (Lewis^a^) were prepared, which were able to strongly bind LecB and allowed higher BF inhibition effects. Finally, in this study, synergistic application of the previously reported dendrimer FD2 with the antibiotic tobramycin at sub-inhibitory concentrations of both compounds was carried out, demonstrating effective BF inhibition and dispersal [171]. The group of Scorciapino, this time headed by Batoni, in 2016, studied linear and dendrimer compounds derived from the previously reported lin-SB056 and den-SB0569 by changing the first two residues [KWKIRVRLSA-NH _2_] of the original sequence [WKKIRVRLSA-NH _2_] and obtained new compounds, namely lin-SB056-1 and den-SB056-1, which possessed an improved amphiphilic profile, and explored the effects of this modification [172]. Den-SB056-1 showed antibacterial effects against both Gram-negative and Gram-positive bacteria, based on the disruption of bacterial membranes. The results obtained against *E. coli* and *S. aureus* planktonic strains confirmed the added value of the dendrimer structure over the linear one, and the impact of the higher amphipathicity on enhancement of the peptide performances [172]. Unfortunately, while SB056 peptides exhibited enthralling antibiofilm properties mainly against sessile cells of *S. epidermidis*, SB056-1 showed reduced antibiofilm effects if compared with that of SB056 [172]. In a study of the same year by VanKoten et al. (2016), a highly cationic fourth-generation PAMAM dendrimer (G4-PAMAM), functionalized with 1-hexadecyl-azoniabicylo [2.2.2] octane (C16-DABCO), a quaternary ammonium compound known to have antibacterial activity, was synthetized [173]. According to the authors, the dendrimer activity depended on both the presence of mannose residues and the ammonium quaternary groups. The PAMAM dendrimer was active against both Gram-positive and Gram-negative species, especially Gram-positive isolates such as *S. aureus* and *B. cereus* [173]. Although the C_16_-DABCO-dendrimer did not show intrinsic antibiofilm effects, it was observed that membranes treated with 1 mg/mL C_16_-DABCO-dendrimer inhibited the formation of *S. aureus* BF, thus establishing its future use in pre-treating membranes and preventing BF formation [173]. Fuentes-Paniagua et al. focused on the antibacterial activity against *S. aureus* and *E. coli* of two types of cationic carbosilane (CBS) dendrimers and dendrons and their hemolytic properties [78]. SAR studies were carried out to evaluate the influence of the generation, type of peripheral groups near the cationic charges, core of dendrimers, and focal point of dendrons (-N_3_, -NH_2_, -OH) on the antibacterial activity of such compounds. These studies evidenced the importance of an adequate balance between the hydrophilic and lipophilic fragments of these molecules. The lowest hemolytic toxicity was registered for dendrimer systems with a sulfur atom close to the surface and when dendrons had a hydroxyl central point. One dendrimer and one dendron, both bearing a sulfur atom close to the surface, scored best in the activity–toxicity relationship analyses, and were chosen for resistance assays [78]. No changes in the inhibitory and bactericidal capacity were observed in the case of the dendron while only a slight increase in these values was noted for the dendrimer after 15 subculture cycles. Furthermore, these two compounds remained active towards different strains of resistant bacteria and prevented the formation of BF at concentrations over the minimum inhibitory concentration (MIC) [78]. In their study in 2019, Barrios-Gumiel and co-workers reported on the preparation of silver nanoparticles (AgNP) covered with cationic CBS and poly (ethylene glycol) (PEG), having antibacterial properties, antifouling effects, and improved biocompatibility due to the presence of PEG [174]. The new family of hetero-functionalized AgNPs was directly synthesized using silver precursor and cationic CBS and PEG ligands containing a thiol moiety. AgNPs were characterized by TEM, TGA, UV, 1H NMR, DLS, Z potential, and XRD. The antibacterial ability of these systems was evaluated against *E. coli* and *S. aureus*, evidencing that the effects obtained for PEGylated systems were slightly lower than those observed for non-PEGylated AgNPs, compensated for by the greater biocompatibility [174]. Furthermore, the authors tested one non-PEGylated and one PEGylated AgNP dendron for their tendency to induce resistance in a planktonic state. Both AgNP-based dendrons barely affected the minimum inhibitory concentration (MIC), whereas the reference antibiotics generated significant resistance [174]. Concerning the topic of the present paper, a relevant improvement in BF inhibition was achieved by dendronized AgNPs after PEGylation [174]. Heredero-Bermejo and Casanova focused on the discovery of dendrimers active against *Candida* spp., which is one of the most common fungal pathogens, the BFs of which, especially those by *C. albicans*, offer resistance mechanisms against most antifungal agents [60]. In their work of 2020, the authors carried out a study concerning the in vitro effects of different cationic CBS dendrimers against BF formation and mature BFs by both Colección Española de Cultivos Tipo (CECT) 1002 and clinical *C. albicans* strains [60]. One out of 14 dendritic molecules tested, BDSQ024 showed the highest activity with a minimum BF inhibitory concentration (MBIC) for BF formation and a minimum BF damaging concentration (MBDC) for existing BF of 16–32 and 16 mg/L, respectively [60]. Additionally, synergistic effects at non-cytotoxic concentrations were detected both with amphotericin (AmB) and caspofungin (CSF) [60]. Subsequently, the research group of Gómez-Casanova et al., using in vitro tests, evaluated the ability of three different generations of cationic CBS dendrons to inhibit the initial formation of *C. albicans* BF and disaggregate the mature BF [175]. Concerning their synthesis, the authors obtained highly water-soluble dendrons (ArCO_2_G_n_(SNMe_3_I)_m_) starting from 4-phenylbutyric acid. The interaction with the cell membranes of the generated systems was found to be dependent on the hydrophilic/lipophilic balance (HLB), which increases with the number of the dendritic generation [175]. Although the compounds showed some toxicity, they showed good antifungal activity against *C. albicans* by inhibiting both the first formation of BF and causing its dispersal [175]. Furthermore, among all the tested dendrons, the second-generation sendron with four positive charges, ArCO_2_G2 (SNMe_3_I)_4_, was found to be the most effective in inhibiting the formation of BF, thus suggesting its hypothetical use as a disinfectant solution for nosocomial surfaces or as a lotion to treat skin infections [175]. Furthermore, ArCO_2_G2 (SNMe_3_I)_4_, when combined with ethylenediaminetetraacetic acid (EDTA) and silver nitrate (AgNO_3_), was able to work on a mature BF, deforming the cell wall and morphology of *C. albicans* [175]. Combination therapy by the association of conventional antibiotics with new compounds represents one of the main strategies used for countering infections by MDR pathogens that produce BFs. Since CBS dendrimers have been proven to be a promising solution for counteracting the formation of BFs, Fernandez et al. developed a new strategy to prevent the formation and/or disaggregation of *S. aureus* BF using a combination of a second-generation cationic CBS dendron with a maleimide group in the focal point and four positive charges MalG_2_(SNHMe_2_Cl)_4_ and levofloxacin (LEV) [176]. With the same purpose, LEV was also combined with a nanoconjugate, formed by a CBS dendron and a cell-penetrating peptide (gH625), called dendron-gH625 nanoconjugate (DPC). The data obtained by Fernandez et al. demonstrated that the combination therapy led to a greater anti-BF effect compared to the concentrations tested individually [176]. In particular, the highest percentages of inhibition were obtained from the combination of DPC-LEV, indicating it is a possible alternative option for BF-associated infections [176]. Moreover, Galdiero et al. (2020) [177] studied the antimicrobial and antibiofilm activities of an analogue of the peptide gH625 (namely gH625-M), a membranotropic peptide described to be a cell-penetrating peptide capable of interacting and disrupting the bilayers of membranes without pore formation. gH625-M was obtained by binding a sequence of lysine residues at the C-terminus, responsible for specific interactions with the negative charges of bacterial membranes [177]. The results obtained by Galdiero et al. showed that the gH625 peptide possessed low antibacterial activity against planktonic cells while it inhibited the initial BF formation of *C. tropicalis, C. serratia, C. marcescens,* and *S. aureus* and was also capable of disintegrating mature BFs formed on silicone surfaces [177]. Furthermore, Galdiero et al., exploiting combination therapy, developed a promising strategy to enhance the efficacy of conventional drugs against BFs by evaluating the possible synergism between gH625-M and commercial antifungal drugs such as amphotericin B (AmphB), fluconazole (FC), echinocandins, and 5 flucytosine (5-FLC) [177]. Among these combinations, gH625-M/FC and gH625-M/5-FLC proved to be effective against BF, unlike single use, even at high concentrations, of antifungals [177]. Very recently, the group of Quintana-Sanchez searched for new microbicide compounds against difficult-to-eradicate BF-forming bacteria, studying cationic multivalent dendrimers both as antibacterial agents and as carriers of active molecules [178]. Particularly, they estimated the antimicrobial activity of cationic CBS dendrimers, unmodified or modified with PEG residues, against planktonic and BF-forming *P. aeruginosa* colonies. This study revealed that the presence of PEG subverted the hydrophilic/hydrophobic balance, and reduced the antibacterial activity, as confirmed by different analytical techniques. On the other hand, the activity was improved by the combination of the CBS dendrimers with endolysin, a bacteriophage-encoded peptidoglycan hydrolase [178]. This enzyme, when used in the absence of the cationic CBS dendrimers, is ineffective against Gram-negative bacteria due to the protective outer membrane shield. On the contrary, the endolysin–CBS dendrimer combination enabled penetration through the membrane and then deterioration of the peptidoglycan layer, providing a synergic antimicrobial effect [178]. To provide readers with an eye-catching full vision of the reviewed scenario, the information reported in the main text is summarized in Table 6.

### 3.3. Promising Areas for Modifying Dendrimer Matrices

As mentioned and confirmed by some case studies reported in the previous section, among cationic dendrimers that are also active on BFs, those with PAMAM, PPI, PEI, poly (lysine), and peptides matrices have been the most extensively prepared and evaluated [63]. Moreover, other structures, such as ammonium-terminated (AT) dendrimers, including AT phosphorous and carbosilane dendrimers, attracted the interest of scientists starting from 2015 [63]. Curiously, even though they were considered to be very attractive for biomedical applications as they are highly biodegradable and have low cytotoxicity, only very recently have polyester-based scaffolds, peripherally cationic for the presence of amino acids, been taken into consideration as novel antimicrobial and or bactericidal devices with very interesting results [4,63,68,69,70,71,72,155]. Regardless, concerning our knowledge, no dendrimer compound of this family has been tested as an antibiofilm agent so far. Particularly, these dendrimers, by presenting an uncharged hydrolysable matrix that is capable of balancing the density of charge on their surface better while retaining strong antibacterial activity, have reduced toxicity [63].

Within this category, peripherally amino acid-modified, polyester-based dendrimer scaffolds, obtained starting from 2,2-bis(hydroxymethyl)propanoic acid (bis-HMPA), as an AB2 monomer, have several features that make them particularly suitable as novel antimicrobial agents. The polyester-based uncharged matrices of these structures, while characterized by good biodegradability, due to the easy physiological hydrolysis, can be esterified with a high number of amino acids, supplying high multivalency, which has been demonstrated to be pivotal for exerting antibiofilm effects. Unfortunately, research in this field is still very limited and, as far as our knowledge is concerned, the literature presents only few studies, without any research on possible antibiofilm effects. In our opinion, this represents a good direction for implementing further studies on the development of new high-generation polyester dendrons and dendrimers peripherally decorated with amino acids to be assessed as antibiofilm agents or assessing the possible antibiofilm activity of those already reported as bactericidal agents.

## 4. Conclusions

Research on the discovery, design, and synthesis of new antimicrobial agents that are also active against BFs and ideally in all phases of its development and/or the development of effective antibiofilm strategies represents the daily challenge of microbiologists, pharmaceutics, organic chemists, pharmacologists, and all scientists who work for global health. The onset of resistance is a phenomenon that affects human health worldwide, but the form of resistance that pathogens develop by BF formation is a dramatic plague against which no current clinically approved drug works. BFs are responsible for the chronicization of BF-associated and biomaterial-associated infections (BAIs), especially those connected with nosocomial and healthcare settings, which translates to increased therapeutic costs and mortality. In this worrying scenario, a recent promising approach to counteract BAIs consists of the use of NAMPs as alternative compounds to existing no-longer-functioning antibiotics due to their mechanism of action as disruptors of the contact of bacterial membranes, which overcome bacterial resistance and limit diffusion. Unfortunately, due to their high costs of production, hemolytic toxicity, and, above all, fast in vivo inactivation by peptidases, their clinical use is still limited. In recent decades, NAMPs have inspired the synthesis of new cationic peptides also in the form of polymers and dendrimer nanoparticles (NPs) that merge the antimicrobial mechanism of NAMPs with the multivalence of polymers and dendrimers, and the nonpareil properties of NPs. While cationic polymer and copolymer NPs have been and are extensively studied for antimicrobial applications, including antibiofilm ones, although the first dendrimers were synthesized in the mid-1980s, and there are several biomedical uses, their use as antibacterial agents, as either drugs, surface coating agents, or drug-delivery systems, has only recently been recognized. In the last decade, dendrimers have mainly been synthesized for the treatment of infections caused by MDR pathogens and some of these have been shown to have an antibiofilm action, but their therapeutic use is still very distant. Here, aiming to incite further development of new antibacterial dendrimers capable of inhibiting the first BF formation and, hopefully, destroying mature BFs, we reviewed the cationic dendrimers developed to this end in the last fifteen years and provided a useful table collecting this information. From this study, it appears that in addition to dendrimer peptides, other forms of cationic dendrimers possess promising antibiofilm activities, being membranotropic molecules that are able to adhere to bacterial membranes and create damage and penetrate the BF biomass. Additionally, the present work evidences that a promising strategy using cationic dendrimers is represented by combination therapy, allowing the combination of conventional drugs with new compounds in order to create synergism following the interaction between the two molecules used. This strategy allows better antibiofilm activity than that obtained with the use of single drugs and the use of lower concentrations of the two ingredients, thus reducing their possible cytotoxicity and, in some cases, increasing the susceptibility of pathogens to antibiotics. We are confident that the extension of the knowledge about these promising but still little-explored materials is a successful approach for discovering new effective weapons for treating chronic infections and biomaterial-associated infections sustained by BF-producing MDR bacteria.

## Figures and Tables

**Figure 1 pharmaceutics-14-02016-f001:**
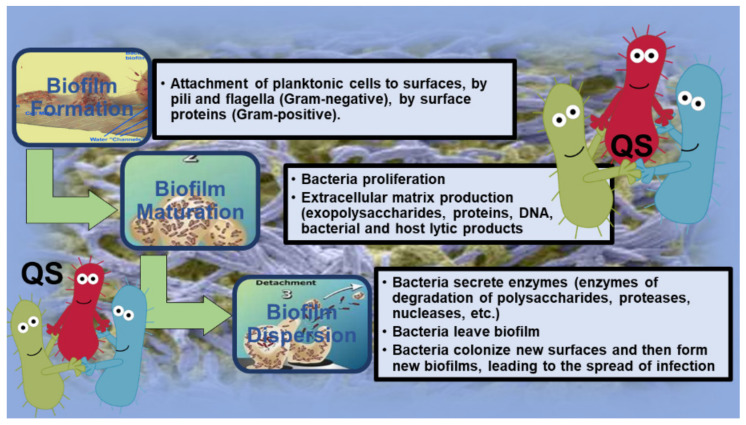
Phases of bacterial BF development.

**Figure 2 pharmaceutics-14-02016-f002:**
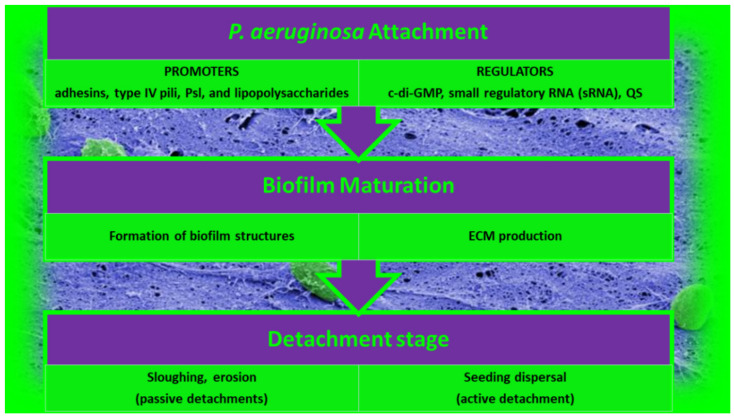
Main developmental stages of *P. aeruginosa* BF. PsL = extracellular polysaccharide expressed by non-mucoid *P. aeruginosa* strains; QS = quorum sensing; ECM = extracellular matrix.

**Figure 3 pharmaceutics-14-02016-f003:**
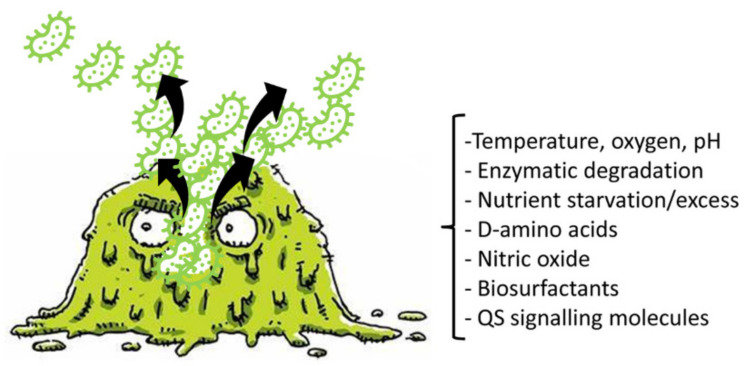
BF detachment.

**Figure 4 pharmaceutics-14-02016-f004:**
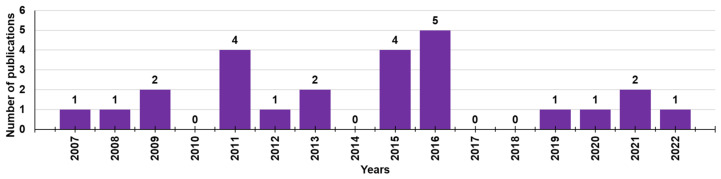
Number of publications per year during the last fifteen years according to Scopus, concerning the development of dendrimers for contrasting BF.

**Table 1 pharmaceutics-14-02016-t001:** Common drug-resistant microbes and diseases caused by them.

Name of Bacterium	Drug(s) Resistant to	Typical Disease
*E. coli*	Cephalosporins, fluoroquinolones	UTI, BSI
*K. pneumoniae*	Cephalosporins, carbapenems	Pneumonia, BSI, UTI
*S. aureus*	Methicillin	Wound, BSI
*S. pneumoniae*	Penicillin	Pneumonia, meningitis, otitis
*Nontyphoidal Salmonella*	Fluoroquinolones	Foodborne diarrhea, BSI
*Shigella* spp.	Fluoroquinolones	Diarrhea *
*N. gonorrhoeae*	Cephalosporins	Gonorrhea
*M. tuberculosis*	Rifampicin, isoniazid, fluoroquinolone	Tuberculosis
**Name of Fungi**		
*Candida* spp.	Fluconazole, echinocandins [7]	Candidiasis
*Cryptococcus neoformans*	Fluconazole [8]	Cryptococcosis
*Aspergillus* spp.	Azoles [9]	Aspergillosis
*Scopulariopsis* spp. *Onychomycosis*	Amphotericin B, flucytosine, azoles [10]	Onychomycosis
**Name of Virus**		
*Cytomegalovirus* (CMV)	Ganciclovir, foscarnet [11]	AIDS and oncology patients
*Herpes simplex virus* (HSV)	Acyclovir, famciclovir, valacyclovir [12]	Herpes simplex
*Human immunodeficiency virus* (HIV)	Antiretroviral drugs [13]	AIDS
*Influenza virus*	Amantadine, rimantadine, neuraminidase inhibitors [14]	Influenza
*Varicella zoster virus*	Acyclovir, valacyclovir [12]	Chicken pox
*Hepatitis B virus* (HBV)	Lamivudine [15]	Hepatitis B
**Name of Parasite**		
*Plasmodia* spp.	Chloroquine, artemisinin, atovaquone [16]	Malaria
*Leishmania* spp.	Pentavalent antimonials, miltefosine paromomycin, amphotericin B [17,18]	Leishmaniasis
*Schistosomes*	Praziquantel, oxamniquine [19,20]	Schistosomiasis
*Entamoeba*	Metronidazole [21]	Amoebiasis
*Trichomonas vaginalis*	Nitroimidazoles [22]	Trichomoniasis
*Toxoplasma gondii*	Artemisinin, atovaquone, sulfadiazine [23,24,25]	Toxoplasmosis

* Bacillary dysentery.

**Table 2 pharmaceutics-14-02016-t002:** General composition of BFs.

BF Biomass	Organized communities of pathogens	Sessile cells	Cells attached to surfaces forming highly coordinated microcolonies, communicating by the QS system, and lacking motility
		Persistent cells	Small subpopulation of microorganisms reversibly transformed into slowly growing cells
		Dormant cells	
	Extracellular polymeric substances (EPSs)	Enzymatic proteins	Polypeptides
		Polysaccharides	Cellulose Polyglucosamine (PGA) Anexopolysaccharides
		Extracellular DNA (eDNA)	
		Cationic and anionic glycoproteins	
		Cationic and anionic glycolipids	Allow real communication between bacteria Stabilize the 3-D structure of BF

**Table 3 pharmaceutics-14-02016-t003:** Important characteristics of *P. aeruginosa* BF.

Functions	Components	Sub Components	Molecules	Function
Matrix Adhesive material Protective barrier	ECMs	EPS	PsL *	Initiation and maintenance of BF Supplies cell surface attachment Supplies intercellular interactions
			PeL *^,^ **	Essential for forming pellicles at the air–liquid interface and solid surface-associated BFs [98,102] Platform for BF structure Supplies protection against aminoglycosides [100,103] Binding with eDNA of the BF [104,105]. Compensates for a lack of PsL in the BF periphery [104]
			Alginate ***	Factor used to distinguish mucoid or non-mucoid *P. aeruginosa* BFs Retains water and nutrients Supplies antibiotic resistance and immune evasion [105,106,107]
		eDNA	Formation of cation gradients Antibiotic resistance Nutrient source Early BF development [99,108,109,110] Major proinflammatory factor for *P. aeruginosa* BFs [111]	
		Proteins	Flagella	Act as an adhesin to help initial bacterial attachment to the surface [112]
			Type IV pili	Contribute to the formation of mushroom-like BF cap structures [113,114]
			CdrA adhesin	Interacts with PsL and increases BF stability [115].
			Cup fimbriae	Important roles in cell-to-cell interaction during the first stage of BF formation [115]
Intercellular communication system enabling bacteria to sense their own population density.	QS system	NAHSL AI-2	las	Regulate several hundred genes in *P. aeruginosa* [116,117] Regulate the bacterial phenotype, spatial differentiation in BFs, motility, and BF formation [118]
			Rhl	
			PQS	
			Integrated QS (IQS)	

* Psl and Pel production occurs through the c-di-GMP signaling pathway owing to environmental signals; ** the Pel production mechanism is implicated in the association of LPS. The complete biochemical composition of Pel has not yet been discovered. Currently, Pel is known to be made of cationic amino sugars; *** mainly produced by strains of *P. aeruginosa* isolated from patients affected by cystic fibrosis (CF); NAHSL = N-acyl-homoserine lactones; AI-2 = autoinducer 2.

**Table 4 pharmaceutics-14-02016-t004:** Bacteria BFs’ contribution in the inactivation of antibiotics and antibiotic treatment failure.

Reasons for the Failure of Antibiotics	BF Function	Factors	Inactivated Antibiotics	Ref.
Hampered antibiotic penetration	Anti-spread barrier	EPS	Ampicillin Ciprofloxacin	[126]
Presence of antibiotic-degrading enzymes	To provide β-lactamases (β-LS)	↑ β-LS	Ampicillin	[127]
			Imipenem Ceftazidime	
Increased BF resistance	To provide eDNA	↑ eDNA ↓ Mg^2+^	Cationic Peptides Aminoglycosides	[128,129,130]
Presence of persistent cells	To cause gradients in nutrients and oxygen concentration To promote differentiation in cell growth	Endogenous stress TA ^1^-systems	Rifampicin Aminoglycosides	[131]
Presence of dormant cells		↓ Functions ↓ Energy ↓ Biosynthesis	Fluoroquinolones	[132]
↑ Resistance to stress	To cause adaptive stress responses by heterogeneity	Changes in component/processes target of antibiotics	Ofloxacin Gentamicin Meropenem Colistin	[133]
			Ofloxacin	[134]
↑ Export of membrane proteins	To up-regulate the production of some efflux pumps	↑ Efflux pumps QS	Multidrugs	[135]
			Azithromycin	[136]
Genetic diversity	To act as a reservoir of genetic diversity by promoting plasmids transfer	Horizontal gene transfer (HGT) eDNA QS	Aminoglycosides	[137]

^1^ TA = toxin/antitoxin; ↑ = improved, higher, increased; ↓ = reduced, decreased.

**Table 5 pharmaceutics-14-02016-t005:** Chemical structure representative of the cationic dendrimers developed in the last fifteen years that were active on pathogens’ BFs.

Dendrimers Structure	Name	Activity
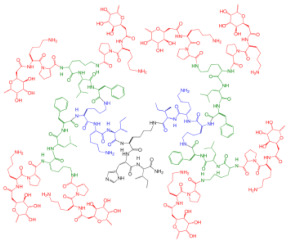	2G3	IC_50_ = 0.025 µM *P. aeruginosa* (PA) LecB
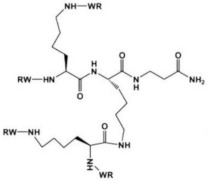	(RW)_4D_	Inactivate *E. coli* RP437 planktonic culture and BFs
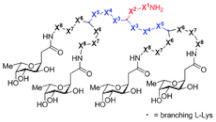	FD2	LD_50_ LecB PA 0.14 µM *P. aeruginosa* BF formation (IC_50_ = 10 mM)
	D-FD2	LD_50_ LecB PA 0.66 µM
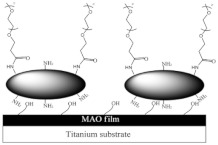	PEGylated PAMAM Film on MAO Substrate	BF by PA (strain PAO1) and *S. aureus* (SA).
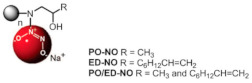	Nitric Oxide-Releasing amphiphilic PAMAM	Inhibited PA BFs
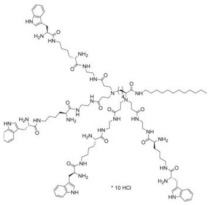	**14**	Inhibited *C. albicans* BF
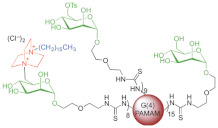	G4 PAMAMs decorated with C_16_-DABCO	Inhibited *E. coli* and *B. cereus* BF
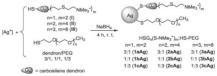	PEGylated AgNPs covered with cationic carbosilane dendrons	Inhibited *E. coli* and *S. aureus* BF

**Table 6 pharmaceutics-14-02016-t006:** Antibiofilm dendrimers developed in the last fifteen years.

Mechanism of Action	Type of Dendrimers	Target Pathogens	Preventing BF Formation	Hindering BF Development	BF Dispersal	Joint Therapy	[Ref.] Year
Inhibition LecB	FD2	*P. aeruginosa*					[156] 2008
2G3					[157] 2009
Membrane (Ms) Disruptors	(RW) _4D_	*E. coli*					[158] 2009
Inhibition LecB	D-FD2	*P. aeruginosa*					[159] 2011
Inhibition LecA	GalAG2 and GalBG2					[160] 2011
Ms disruptors	(RW)_4_-NH_2_	*E. coli*				20 µM +0.5 µg/mL * 0% BF cells	[161] 2011
(RW) _4D_	*E. coli* HM22				20 µM +0.5 µg/mL * <10% BF cells
Ti-S(CaPO_4_)060-PEGPAMAMs Ti-S-060-PEGPAMAMs	*P. aeruginosa* *S. aureus*					[162] 2011
SB056	*S. epidermidis P. aeruginosa*					[163] 2012
Ms disruptors Release of NO	NO-releasing PAMAMs #	*P. aeruginosa*					[164] 2013
Inhibition LecA Inhibition LecB	TGPDs	* P. aeruginosa *					[165] 2013
Membranolytic Apoptotic	Cationic antifungal peptide dendrimers (**9**, **14**)	* C. albicans *					[166] 2015
Outer cell wall damage Absence of true hyphae Filamentation inhibition Ms permeabilization	AgNPs	* C. albicans *					[59] 2015
Outer cell wall damage Ms permeabilization BF mass penetration	2D-24	* P. aeruginosa *				Ciprofloxacin (Cip) ^1^ Tobramycin (Tob) **^, 1^ Carbenicillin (Car) ^1^	[152] 2015
Ms disruptors Release of NO	(G1–G4)-NO-releasing alkyl-PAMAMs #	*P. aeruginosa**S. aureus*MRSA					[167] 2015
Electrostatic interactions Ms disruptors Release of NO	(G1)-NO-releasing alkyl PAMAMs #	*S. mutans*					[169] 2016
Inhibition LecA	G3/G4 analogous of GalAG2 and GalBG2	* P. aeruginosa *					[170] 2016
Inhibition LecA Inhibition LecB	Het1G2-Het8G2 Het1G1Cys-Het8G1Cys Others	* P. aeruginosa *				Dendrimer FD2+Tob	[171] 2016
Ms disruptors	den-SB056-1	* S. epidermidis *	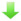	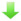	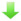		[172] 2016
C_16_-DABCO-G4-PAMAM	* S. aureus * *B. cereus*	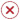 §	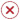	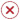		[173] 2016
NH_4_^+^ carbosilane dendrimers and dendrons (CBSD)	* S. aureus * CECT240					[78] 2016
Ms disruptor Ag release	PEGylated-NH_4_^+^ CBSD-AgNPs	* S. aureus * *E. coli*					[174] 2019
Ms disruptor	BDSQ024	* C. albicans *				Amphotericin (AmphB) Caspofungin (CSF)	[60] 2020
ArCO_2_G2 (SNMe_3_I)_4_	* C. albicans *			 ^2^	Ethylenediaminetetraacetic acid (EDTA) + silver nitrate (AgNO_3_)	[175] 2021
Interaction/disruption of the Ms bilayers	MalG2(SNHMe _2_Cl)_4_ DPC	*S. aureus*	 ^2^	 ^2^		Levofloxacin (LEV)	[176] 2021
Interaction/disruption of the Ms bilayers No pore formation	gH625-M ^3^	*C. tropicalis* *C. serratia* *C. marcescens* *S. aureus*				AmphB ^4^ Fluconazole (FC) Echinocandins ^4^ 5 Flucytosine (5-FLC)	[177] 2020
Interaction/disruption of the Ms bilayers	PEG/no-PEG cationic CBSD	*P. aeruginosa*	 ^2^	 ^2^	 ^2^	Endolysin	[178] 2022

* ofloxacin; ** best association; ^1^ target the synthesis of DNA, proteins, and cell wall; # NO payloads of ~1.0 μmol/mg; 
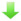
 = reduced with respect to SB056; § the treatment of membranes with 1 mg/mL causes complete inhibition of the growth of *S. aureus* BFs; ^2^ when in combination as in the column 7; ^3^ analogous of the cell penetratin peptide gH625; ^4^ not functioning; HM22 denotes a strain containing the *hipA7* allele, which maps to the *hipA* gene in the antitoxin-toxin module HipBA. The expression of the *hipA7* allele confers a 1000-fold higher frequency of persister cells formation; AgNPs = silver nanoparticles; MRSA = methicillin-resistant *S. aureus*; 

 = not reported; 

 = yes, active; 
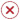
 = no, not active.

## Data Availability

Not applicable.

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
