# Peer review of "Prevention and Eradication of Biofilm by Dendrimers: A Possibility Still Little Explored"

_pharmaceutics, 2022, doi:10.3390/pharmaceutics14102016_

Round 1

Reviewer 1 Report

This paper gives a comprehensive overview of dendritic biocides, but the focus is so strong that some important issues are omitted. For instance, nothing is mentioned on the way how to use these compounds and how do they are when compared with other close related compounds. Although it is mentioned that it is a disadvantage that natural peptide are degraded in the body, this is neglected in the dendritic compounds

Author Response

This paper gives a comprehensive overview of dendritic biocides, but the focus is so strong that some important issues are omitted. For instance, nothing is mentioned on the way how to use these compounds and how do they are when compared with other close related compounds. Although it is mentioned that it is a disadvantage that natural peptide are degraded in the body, this is neglected in the dendritic compounds.

We thank the Reviewer for his comments. We hope that following his suggestions detailed below we will able to address all his concerns.

General remark.

The biological part, which is not strictly my aera of expertise, is well written and gives a good overview of various aspects of biofilms. Although many details are given for specialists, it also gives a good background information for specialists and for those who are active in nearby research topics. It is a comprehensive overview of dendritic, peptide- containing biocides

We thank a lot the Reviewer for his appreciations.

In the second part the authors focus this manuscript on dendrimers, expensive, but perfect polyamines, but they don’t mention e.g. branched polyethyleneimine (PEI). PEI is a less perfect polyamine than polypropylene imine dendrimer, but has similar properties as dendrimers and is much cheaper. Moreover, the special advantage of using dendrimers is hardly mentioned.

We thank the Reviewer for his comment. We agree with him. So, a paragraph concerning PEI has been included in Section 3.1.1. (lines 439-449), while the advantages associated to the use of cationic material having dendritic structure was more in deep discussed in the same Section 3.1.1.(lines 457-461).

In the introduction the authors are sceptical over the use of natural peptides, as they degrade shortly after introduced in the body. Moreover, they are very expensive. In contrast they are very enthusiastic over peptide-dendrimers, which will probably suffer by the same problems.

We thank a lot the Reviewer for his appreciable comments. However, we make kindly note to the Reviewer, that we were not skeptical concerning the use of natural peptides, which, even if expensive and susceptible of early degradation, are well functioning, as well as we have not expressed particular enthusiasm for peptide-dendrimers which surely could suffer by the same problems. On the contrary, we have introduced other type of dendrimers, less toxic than PAMAM and more stable than peptide dendrimers, which among other have shown potent bactericidal effects, and about which we are enthusiastic. Please, see lines 449-453. Regarding the case studies reported by us, they are all the developed ones in the last fifteen years, reporting of materials active on the biofilm (topic of this review). Many concern the synthesis and antibiofilm effects of peptide-dendrimer, but this fact does not depend on our decisions.      

They authors pay no attention to the way the biocides could be applied. Is the review only on antibacterial compounds, no matter how they are applied? Only in vitro experiments are mentioned. Are the compounds additives, coatings (immobilized or leachables) or construction materials? The substrate and the way biocides are applied on a substrate plays an important role, but is not mentioned. Only on page 17 (line 502) they mention the formation of dendrimer films. It would be welcome to give some information on how could these materials could be used?

As mentioned in the main text, the studies on antibacterial/bactericidal cationic dendrimers active also on biofilms are limited and still at very early stages, and only 29 works reporting in vitro experiments are available. Concerning how these materials can be used, some information was already present in the original version of the Review, anyway, to satisfy the Reviewer, additional information has been included for the case studies reported, as available.

Page 4, line 112. Typing error SQ should be QS.

c compounds.

We apologise with the Reviewer for our distraction. The typing error has been corrected (line 115 of the revised version).

Reviewer 2 Report

The review titled “Prevention and eradication of biofilm by dendrimers: A possibility still little explored” is devoted to the problem of multi-drug resistance, which is increasingly emerging in various human pathogens (bacteria, fungi) toward antibiotics. The appearance multi-drug resistance makes it difficult to choose the right treatment regimen, slows down the recovery of patients, and in some cases leads to extremely serious consequences. In nature, antifilm properties are exhibited by some antimicrobial peptides, by analogy with which some cationic polymers and dendrimers can act. The present review is devoted to a discussion of works over the past 15 years, which provide information on such synthetic antifilm compounds. The review is well written. The theme corresponds to the scope of the Pharmaceutics journal. Meanwhile there are some points that need to be revised before this manuscript can be recommended for publication.

 Comments:

 1. The proportion of references over the past 5 years is relatively small. Does this mean that interest in this problem is declining? If possible, add references.

 2. Table 2 needs to be restructured. The word “Biomolecules” should probably be completely removed from the table. Authors should decide on what information they emphasize: on the function performed or examples of substances?

 3. Paragraph 2.1.2.: Describe in more detail the process of maturation of P. aeruginosa bacterial cells.

 4. The last part of the review is devoted to a detailed study of dendrimers used in recent decades. Illustrative material including the effect of dendrimer structure (size, charge, presence of functional groups) on antimicrobial activity may improve this part of the review.

 5. Structural formulas of the most promising dendrimers for biofilm destruction should be given.

 6. Highlight a separate section at the end of the review on promising areas for modifying the dendrimer matrix.

Author Response

The review titled “Prevention and eradication of biofilm by dendrimers: A possibility still little explored” is devoted to the problem of multi-drug resistance, which is increasingly emerging in various human pathogens (bacteria, fungi) toward antibiotics. The appearance multi-drug resistance makes it difficult to choose the right treatment regimen, slows down the recovery of patients, and in some cases leads to extremely serious consequences. In nature, antifilm properties are exhibited by some antimicrobial peptides, by analogy with which some cationic polymers and dendrimers can act. The present review is devoted to a discussion of works over the past 15 years, which provide information on such synthetic antifilm compounds. The review is well written. The theme corresponds to the scope of the Pharmaceutics journal. Meanwhile there are some points that need to be revised before this manuscript can be recommended for publication.

 Comments:

  1. The proportion of references over the past 5 years is relatively small. Does this mean that interest in this problem is declining? If possible, add references.

We thank the Reviewer for paying attention to this data. The reasons proposed by us for this worrying phenomenon have been added to the lines 376-382.

  1. Table 2 needs to be restructured. The word “Biomolecules” should probably be completely removed from the table. Authors should decide on what information they emphasize: on the function performed or examples of substances?

The Reviewer is right. The Table 2 has been restructured removing “biomolecules”.

  1. Paragraph 2.1.2.: Describe in more detail the process of maturation of P. aeruginosa bacterial cells.

As requested by the Reviewer the processes of maturation of BF by P. aeruginosa has been better described by adding mor details (Section 2.1.2.). Additionally, the same improvement was applied also at section 2.1.3.

  1. The last part of the review is devoted to a detailed study of dendrimers used in recent decades. Illustrative material including the effect of dendrimer structure (size, charge, presence of functional groups) on antimicrobial activity may improve this part of the review.

We agree with the Reviewer, but the information required by him has been already inserted when available.

  1. Structural formulas of the most promising dendrimers for biofilm destruction should be given.

As requested by the Reviewer, the structural formulas of the most promising dendrimers for biofilm destruction have been given, inserting them in the new Table 5.

  1. Highlight a separate section at the end of the review on promising areas for modifying the dendrimer matrix.

We thank the Reviewer for this relevant suggestion. We have inserted the requested information in the new sections 3.3. (lines 813-840). We thank the Reviewer for his helpful revision work. We have addressed all the issues raised by him.  

Round 2

Reviewer 2 Report

Authors took into account the comments of the reviewer and made corrections to the text of the manuscript. In this form, it can be recommended for publication.